# Acceptability of test and treat with doxycycline against Onchocerciasis in an area of persistent transmission in Massangam Health District, Cameroon

**Rogers Nditanchou** [1]*, **Ruth Dixon**[2], **Kareen Atekem**[1], **Serge Akongo**[1], **Benjamin Biholong**[3], **Franklin Ayisi**[3], **Philippe Nwane**[4], **Aude Wilhelm**[2], **Sapana Basnet**[2], **Richard Selby**[2], **Samuel Wanji**[5,6], **Didier Bakajika**[7], **Joseph Oye**[1], **Joseph Kamgno**[4,8], **Daniel Boakye**[9], **Elena Schmidt**[2], **Laura Senyonjo**[2]

**1** Sightsavers, Cameroon Country Office, Cameroon, **2** Sightsavers, Haywards Heath, United Kingdom, **3** National Programme for the Fight against Onchocerciasis and Lymphatic Filariasis, Ministry of Public Health, Yaoundé, Cameroon, **4** Filariasis and other Tropical Neglected Diseases Research Center, Yaoundé, Cameroon, **5** Research Foundation in Tropical Diseases and Environment, Buea, Cameroon, **6** Department of Microbiology and Parasitology, University of Buea, Cameroon, **7** WHO/Regional Office for Africa, Brazzaville, Republic of the Congo, **8** Faculty of Medicine and Biomedical Sciences, University of Yaoundé I, Yaoundé, Cameroon, **9** Parasitology Department, Noguchi Memorial Institute for Medical Research, University of Ghana, Accra, Ghana

* rnditanchou@sightsavers.org

**Data Availability Statement:** All relevant data are within the manuscript and its Supporting Information files.

## Abstract

The main onchocerciasis elimination strategy is annual Community-Directed Treatment with ivermectin (CDTi). However, as a response to persistent high infection prevalence in Massangam Health District in Cameroon, two rounds of alternative treatments including biannual CDTi, ground larviciding and test and treat with doxycycline (TTd) were implemented. This led to a significant prevalence reduction from 35.7% to 12.3% (p<0.001) as reported by Atekem and colleagues. Here we report on the acceptability of TTd component based on qualitative and quantitative data. The TTd involved microscopic examination for microfilaria in skin biopsy and those infected were offered doxycycline 100 mg daily for 35 days by community-directed distributors (CDDs). Participation level was significantly high with 54% of eligible population (age > 8, not pregnant, not breastfeeding, not severely ill,) participating in the test in each round, increasing to 83% over the two rounds. Factors associated with non-participation included mistrust, being female; being younger than 26 years; short stay in the community; and belonging to semi-nomadic sub population due to their remote and disperse settlement, discrimination, their non selection as CDD, and language and cultural barriers. Treatment coverage was high -71% in round 1 and 83% in round 2. People moving away between testing and treatment impacted treatment coverage. Some participants noted mismatch between symptoms and test result; and that ivermectin is better than doxycycline, while others favoured doxycycline. CDD worried about work burden with unmatching compensation. Overall, TTd participation was satisfactory. But can be improved through reinforcing sensitisation, reducing time between test and treatment; combining TTd and CDTi in one outing; augmenting CDDs compensation and/or weekly visit; exploring for frequently

**Funding:** This work is funded by public donations to Sightsavers. Employees of Sightsavers played pivotal roles in study design, data collection and analysis, the decision to publish, and preparation of the manuscript.

**Competing interests:** The authors have declared that no competing interests exist.

excluded populations and adapting strategies to reach them; and use of a sensitive less invasive test.

## Author summary

Doxycycline was offered to people to treat river blindness in Massangam health district after test indicating they are infected. The test, conducted twice, involved taking a small piece of skin for examination. Those infected were given doxycycline for 35 days by community volunteers. Among those who were eligible for the test, 54% were tested each time, increasing to 83% for both rounds. Participation was influenced by trust level; and varied by sex, age, and duration of community stay. Semi-nomadic population participated less than permanent residents because of their remote locations, language and cultural barriers, perceived discrimination by the permanent resident population. Doing test before ivermectin treatment impacted test result. Delay in starting treatment led to some infected people moving away. Almost all those starting treatment completed it each time. Some beneficiaries prefer the usual ivermectin treatment. For the volunteers, delivering treatment was demanding in terms of effort and time. Overall, the level of participation was satisfactory but could be improved through more sensitisation, reviewing volunteer compensation, inclusion of the frequently excluded population, combining doxycycline and ivermectin treatments, and developing a sensitive less invasive test.

## Introduction

Onchocerciasis is a parasitic disease that causes skin lesions, severe itching, visual impairment and irreversible blindness [1]. The main approach for control and elimination of onchocerciasis is interruption of transmission through annual Community-Directed Treatment with microfilaricide, ivermectin (CDTi). Although significant progress has been made in its control, there are still considerable challenges on the road to elimination due to very high initial or persistent transmission, starting treatment late or co-endemicity with *Loa loa* (a contraindication for ivermectin treatment) among others [2]. In these situations, complementary or alternative treatment strategies (ATS) may be required to meet elimination targets. ATS are all strategies other than the usual annual CDTi and include optimised CDTi, biannual or pluriannual CDTi, complementary vector control, test and treat (T&T) and the use of alternative drugs to ivermectin, including doxycycline [3, 4].

Doxycycline is a macrofilaricide and sterilising antibiotics acting by killing the intracellular endosymbiont *Wolbachia* bacteria living within adult and microfilariae. Doxycycline 100mg given daily for five to six weeks is curative as it completely and irreversible deplete *Wolbachia* leading to the death of adult worms [5–8]. Modelling has demonstrated that doxycycline can deplete *Wolbachia* in 91%-94% of *Wolbachia*-positive adult worms; thus, reducing from 10 years to 2–3 years the lifespan of the worms. This effect is enhanced when combined with ivermectin [9, 10].

Doxycycline is an important alternative treatment to ivermectin. In suspected ivermectin resistant individuals reported to still harbour *Onchocerca volvulus* mf despite ivermectin treatment, doxycycline at a dose of 100mg/day was offered for six weeks. At 20 months post-treatment follow-up, only 5.1% of female worms from treated individuals compared to 80% for the non-treated, harbours *Wolbachia*. All onchocercal nodules removed from the treated patients

were without microfilariae and almost all of the patient (97%) has no microfilaridermia [11]. Similar outcome has been obtained when doxycycline is implemented in onchocerciasis—loa-loa co-endemic settings [12]. In two Cameroonian Health Districts, doxycycline was successfully delivered (without prior test) for six weeks to 73% of 17519 eligible population with 97.5% treatment adherence. Four years later, mf prevalence was found to be lower (among those treated with a combination of ivermectin and doxycycline compared to those who took only ivermectin (17% vs 27.0% p = 0.014) [13, 14]. This shows evidence of impact and feasibility of community-based treatment with doxycycline.

Massangam Health District (HD) in the West Region of Cameroon is one such place where ATS is required. Findings in 2016 showed infection level by skin snip microscopy as high as 37.1% in Makouopsap community [15]. This suggest persistent ongoing transmission and that the area is not on track to elimination despite nearly 20 years of uninterrupted annual CDTi with reported achievement of minimum programmatic coverage [16] (>80% therapeutic coverage) [15, 17]. Responding to the high infection, we implemented ATS that included biannual CDTi, ground larviciding and test and treat with doxycycline (TTd) [18–20]. Unlike in the COUNTDOWN project [21] where each strategy was implemented separately, in Massangam all the strategies were delivered to the same focus/area (Fig 1). CDTi and TTd were delivered in the high focal community of Makouopsap, Makakoun and Njijia/Njigouet, all bordering each other, and ground larviciding conducted in rivers in and around the focal communities where black flies (vector of onchocerciasis parasite) breeding sites have been previously detected [16]. An impact evaluation of these interventions showed a significant and rapid reductions in mf prevalence (from 35.7% to 12.3%; p<0.001) and Community Microfilaria Load (CMFL), from 0.66 to 0.22 mf/ss in the focal communities over a two-year period. Detail report of the impact (effectiveness) of the whole ATS package is presented elsewhere [18]. This paper focuses on the acceptability of the TTd component.

To date, TTd delivery has primarily been within research settings. This implementation in the focus of Massangam was delivered in a programmatic setting. Exploration of its acceptability, and lessons learnt during operationalisation are essential for replication in foci encountering difficulties reaching elimination. We report findings from field operations, meeting notes, partner activity reports and qualitative assessment of the TTd. In addition, we reflect on barriers and enablers for implementation and report the solutions and adaptations developed before concluding with recommendations for future implementation of TTd.

## Methods

### Ethics statement

Ethical clearance for this study was obtained from the *Comité National d'Ethique de la Recherche pour la Santé Humaine* (CNERSH) in Yaoundé, Cameroon (clearance N° 2017/06/918/CE/CNERSH/SP). Written informed consent was acquired ahead of participation in any aspect of the project, as was parental consent (written) for any individual aged 17 or under.

Test and Treat with doxycycline (TTd) was conducted over two rounds, the first in July 2017 (intervention 1) and the second round in September 2018 (intervention 2). These interventions were supported by field monitoring and detailed notes. In addition, partners on the field provided reports and meeting minutes summarizing encountered challenges and adjustments made during the implementation (See S1–S3 Texts). A qualitative evaluation was conducted in November 2018 after round 2 TTd. The timeline of the activities is found in supplementary materials (S1 Table). TTd steps included mobilisation and census, enrolment, skin snip test and treatment.

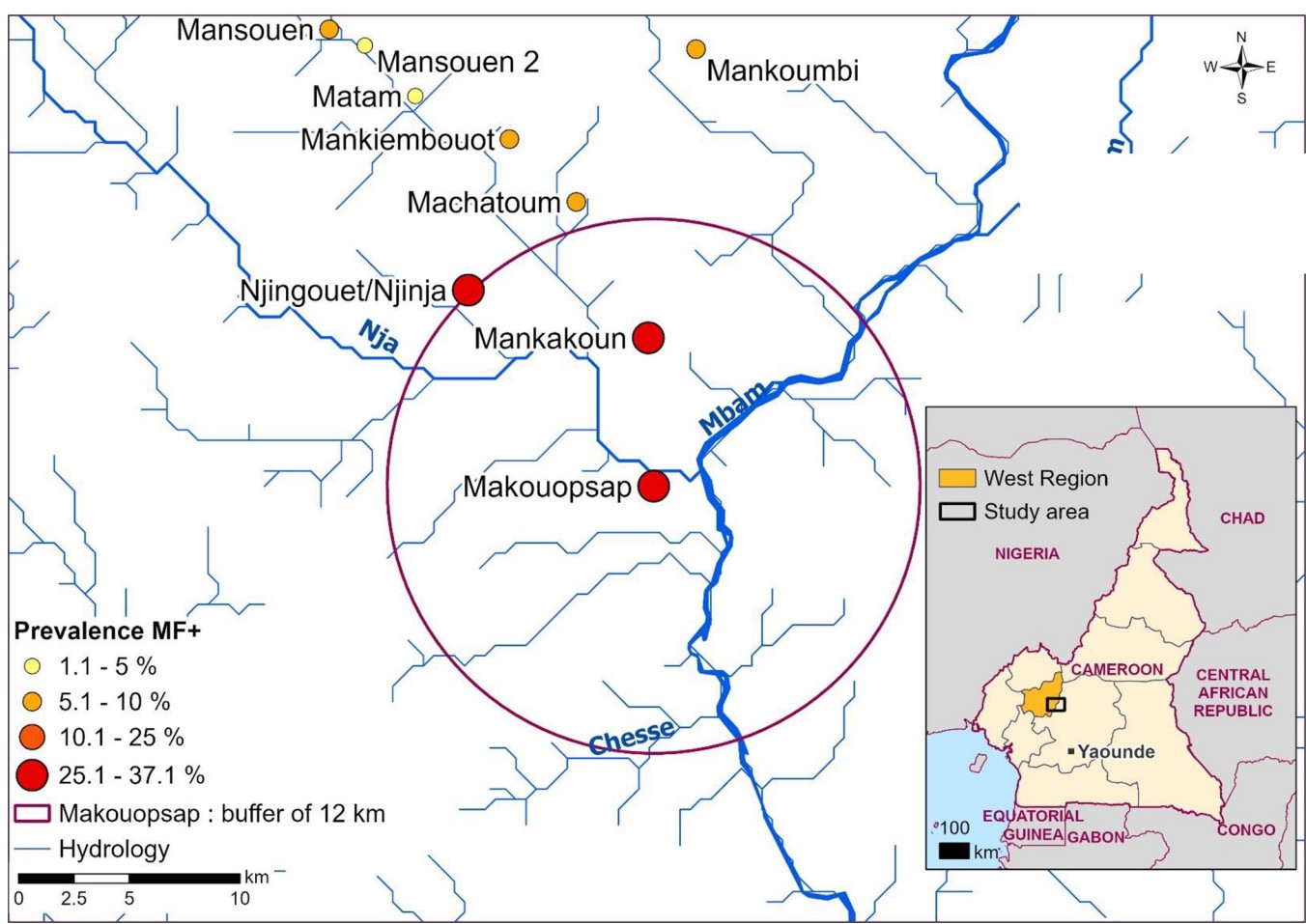

**Fig 1. Map of Massangam Health Area showing area of intervention (circle).** Circle indicates high transmission foci communities (high transmission focus) where doxycycline and ivermectin treatment were implemented [18]. In the rest of the communities, only biannual ivermectin mass drug administration was implemented. Larviciding was implemented in Njah and Mbam rivers following baseline survey that identified breeding sites contributing to transmission in the area [15]. (Sources: Hydrology-HydroSHEDS https://www.hydrosheds.org/; the administrative boundaries—Natural Earth https://www.naturalearthdata.com/downloads/; all Community locations were collected during the field activity by using GPS devices. Map was created using ArcGIS PRO® software by Esri).

### Mobilisation and census

Mobilisation and sensitization included various activities to raise community awareness and participation in ATS. A launching ceremony was held at the district level. Participants in this meeting were community and religious leaders, community members, local authorities, and programme staff at the district health service. Following launching, smaller meetings with community members were organised in each community to emphasize the importance of the project, understand concerns and potential challenges that may be faced during implementation, and understand the most effective strategies to encourage participation of the community members. These meetings were organised in a more informal fashion to facilitate open dialogue with community members and encourage them to ask questions. Sensitisation tools used included posters pasted at public places such as market squares, mosques and churches, schools, and other accessible spots; flyers, distributed to those who participated in the meetings and during each activity; banners hoisted at market squares, community leader residents and health facilities; megaphone use by town crier to announce the activity and invite people to participate; and announcement in local radio in local languages/dialects—Bamoun, Fulfulde,

English and French languages. A mobile caravan mounted with musical instrument played round the community was spectacular in drawing attention to the announcement being made. Also, community leaders swallowed drug in public to assure and motivate others. Furthermore, efforts were made to include female as CDDs.

Following mobilization and sensitization, a census was conducted prior to screening by two teams of trained surveyor assisted by community researchers. It was a house-to-house census in which the team visited every household and recorded individual name, age, sex, occupation, tribe, duration of stay in the community and participation in previous TTd (during intervention 2). This information was collected using EpiCollect 5 software (https://five.epicollect.net/) installed on android smartphones in round 1. In the 2nd round, census was recorded in paper-based register and later entered into Excel spread sheet. During the census, sensitization was reinforced by distributing flyers, pasting posters, and informing the people of the activities, its benefits and inviting them to participate.

### Enrolment

After obtaining written informed consent and additional ascent for those younger than 18 years, eligible individuals—above the age of 8, not pregnant, not breast feeding and not having severe illness, not allergic to doxycycline—were invited to a fixed location for onchocerciasis screening (test) through skin snip biopsy [18]. Pregnancy was excluded by performing urinary dipstick test for all women of child-bearing age (>15 years) and only those showing a negative test were included. In addition, severe illness was excluded by a medical doctor who examined every participant, evaluating for hypertension (measuring blood pressure) and diabetes (using rapid test). As for census, enrolment, and skin snip testing in round 2 was irrespective of enrolment in round 1.

### Skin snip test

After obtaining consent and ascertaining eligibility during enrolment, skin snip test was conducted. The procedure of skin snip test consisted of taking two bloodless skin snips from each iliac crest of an individual using a sterile corneo-scleral punch. The snips were then put into wells of a 96 well-micro-titre plate in which two drops of saline solution had been previously placed and sealed with parafilm to avoid spillage and evaporation. It was then allowed to incubate for 24 hours at room temperature. The incubation medium was examined under a compound microscope (x10) for the presence (infected) or absence of *O. volvulus* mf. Microscopic examination was conducted in one field station during round 1. In round 2, this exam was done in each community and allowing community members to view mf under the microscope when found. This was intended to reinforced trust and participation as an adaptation from round 1. Another adaptation during round 2 was the immediate return of the results. Infected participants were invited to community health facilities for their results. Those absent were contacted by phone; and failingly, by visiting them at their houses. The results were returned individually in a written note contained in a sealed envelope.

### Doxycycline treatment

If infected and eligible (above the age of 8, not pregnant, not breast feeding and not having severe illness), participants were invited for treatment through a community-based approach. They were first educated on the duration, the need to adhere to treatment, eat before treatment to avoid side effect, timing of treatment, a place to gather for treatment, reporting side effects and obtaining consent. The infected individuals were grouped by proximity to a CDD who offered them light meal before they swallowed doxycycline 100mg tablet or capsule

(Vibramycin, Pfizer in the 1st round; generic doxycycline in 2nd round) every day under the watch of the CDD (directly observed therapy (DOT) strategy) for 35 days. Treatments were monitored for completion and side effects—diarrhoea, nausea, vomiting, headache, itches, skin eruption or others (detailing nature, severity, and frequency) daily by CDDs. This was done through questionnaire or beneficiary complaint. Treatments were noted in treatment register and side effects were noted by ticking or shading the corresponding option (s) in treatment register (See S2 Table).

## Data management and analysis

Screening and treatment data were recorded on paper-based registers and then entered into Excel and imported into Stata statistical analysis software where it was cleaned and analysed. We acknowledged that manual data record can be vulnerable to error as it entails data entry to Excel sheets from paper, which is prone to human error. To minimize this, two individuals entered the same data which were merged, and any discrepancy verified and corrected. Individuals were tracked between interventions by name, sex, age, and community. Test participation was calculated as a percentage of eligible population participating in the test. Pearson Chi-square test was used to compare proportion of skin snip test participation between rounds 1 and 2. A cumulative screening participation was calculated as the number of individuals participating in one or both interventions over the registered eligible population recorded during both rounds. Infection percentage (prevalence) was determined as percentage of those infected. Treatment start rate and course completion (rate) was calculated as percentage of those starting and completing treatment, respectively.

A multiple logistic regression model was used to compare the screening participation, the positivity proportion, the treatment starts and the proportion of participants completing the treatment course in both interventions. This analysis was refined by considering subgroups such as sex, age, community, length of stay in the community, and tribe. In checking for the suitability for modelling, there was a limitation that some cells have fewer than 10 events. However, the number of cells with <10 events were very few–two. Analysis was then refined by considering gender, age-category, community in a multiple logistic regression modelling (MLR) to obtain adjusted OR. None of the explanatory variables has correlation coefficient >0.3 between each other. So, we considered noncollinearity among the variables. The adjusted OR and fitness of the regression model were reported alongside their p-values. Fitness of the model was estimated using LR $\chi2$ (Likelihood ratio Chi-square test) and corresponding p-values ($p<0.05$ is considered statistically significant).

## Meeting minutes and field notes

In 2016, the high prevalence situation of onchocerciasis in Massangam HD [16] and the proposed ATS solutions were presented and discussed in an advocacy meeting attended by the Ministry of Health (programme), technical experts and Neglected Tropical Diseases (NTD) implementing partners including Sightsavers. In this meeting, ATS was endorsed for implementation in Massangam. At the start of each TTd round, a meeting involving implementation team and technical experts was held to review progress and challenges and agree on adaptations/solutions. In addition, field monitoring was conducted, and field observations, challenges and adaptations were noted. The details of these meeting reports and minutes are found in supplementary materials (See S1–S3 Texts).

## Qualitative assessment

Seventeen Focus Group Discussions (FGD) and 30 Key Informant Interviews (KII) were conducted using interview guides to explore participants' experiences of TTd. The number of KII were determined following the organisation of the National Programme to include key persons at the regional, district and community levels: Regional—Regional NTD focal person (1); District: district medical officer (2), health area chief (1) and head of health facilities (3); Community (3 –Makouopsap, Makakoun and Njijia/Njigouet): CDDs (5), opinion leaders for settled (8) and semi-nomadic population (2) and community members (semi-nomadic 5; settled 5). The number of FGD groups were determined to reflect the three communities and the sub-population—settled and semi-nomadic and within each sub-population constitute groups of 6–10 CDDs, youths, women, men (from far and near section of the communities) for the group discussion. They were recruited through the community dialogue structure by inviting volunteers from the groups. Table 1 shows the number of KIIs and FGDs following the health district and community organisation.

Previous study identified semi-nomads as not equitably reached during ivermectin Mass Drug Administration (MDA) [22]. Nomads are members of traditional groups of hunter-gatherers and pastoralists who do not have a fixed habitation and regularly move locations. When there is a base to which the nomads return seasonally, their lifestyle is termed semi-nomadic [23]. They often face inequitable access to health interventions. Semi-nomads in Massangam area generally come from the northern part of Cameroon, Nigeria, and Chad.

KII and FGD guides were developed for different subgroup including Ministry of Health (programme staff, from the region to the community health facility levels), Community-Directed Drug Distributors, and the communities–both settled and nomadic (see S4 Text). The guides were used to ask and probe for responses on awareness, burden, and treatment of onchocerciasis. In addition, it included questions and probes to explore participation, barriers, motivators, perception, and suggestions of how TTd could be improved. Both FGD and KII interviews were conducted at the participants resident or as close to as possible.

KII and FGD were conducted by 3 teams of two (one interviewer and an assistant) in the interviewee's language. Each interview (KII and FGD) lasted between 30 minutes and one and a half hour. The teams tape-recorded the interviews and took field notes which they reviewed at the end of each day to determine completeness and debt of responses/saturation and need for further probing. The number of FGD (17) and KII (17) were adequate to explore the topic of interest to saturation level. Participation was voluntary. All participants signed informed

**Table 1. Summary of qualitative evaluation participants.**

| Participants | | KIIs[1] (individuals) | #FGDs[2] (group discussion) |
|---|---|---|---|
| Programme personnel | | 5 | - |
| Community Drug Distributors | Semi-nomadic | 0 | - |
| | Settled | 5 | 1 |
| Community and opinion leaders | Semi-nomadic | 2 | - |
| | Settled | 8 | - |
| Community Members | Semi-nomadic | 5 | 7 |
| | Settled | 5 | 9 |
| **Total** | | **30** | **17** |

[1]Key informant Interviews

[2]Focus Group Discussion. The numbers indicate the number of interviews–for Key Informant Interviews (KII) and Focus Group Discussion (FGD) for the settled and semi-nomadic subgroups.

consent and assent for those younger than 18, before participation. Each participant was compensated with a cube of soap costing 250 Central African Francs CFA (about 0.39 United State Dollars).

Interviews records were transcribed verbatim. Transcripts were imported into Nvivo qualitative data analysis software (Version 12, QSR International Pty Ltd; www.qsrinternational.com/nvivo). They were then coded into themes and subthemes by a trained researcher and reviewed independently by two qualitative assessment experts. Details of the coding are found in (see S3 Table). Findings are presented alongside field notes/reports, meeting minutes, and analysis of census, skin snip test and treatment data. Appropriately de-identified quotes were extracted to support the themes.

## Results

### Participation in skin snip test

Skin snip test participation levels during intervention rounds 1 and 2 are shown in Fig 2 and detailed in Table 2. In intervention 2, more residents were registered (2,380) and eligible (1,615) than in intervention 1, 1921 registered and 1,211 eligible (>8 years, not pregnant, not breastfeeding, not severely ill) (Table 2). Combining both intervention rounds together in a matched dataset, in total, 3,549 individuals were registered by census. From the 3,549 total: 1,169 registered only in intervention 1 and were not re-registered during intervention 2; 1,628 registered only in intervention 2; 752 individuals were registered in both rounds. Thus, an additional 1628 (46% of combined population) were identified in the second intervention round. Based on this intervention 1 and 2 matched data, a total of 2,503 (70.5%) were eligible age-wise for at least one of the 2 interventions and 1,433 of them were tested at least once. This equates to a cumulative skin snip testing participation level of 57.3% (1,433) of the age-eligible population. However, considering self-reported participation in round 2 when asked whether they had participated in round 1 skin snip test, 83% (1,337) of age-eligible population participated in the skin snip test in at least one round. It should be noted that all censused, and eligible population were invited for skin snip testing during round 2 irrespective of their participation during round 1. Seven percent (167) of all registered during intervention 2 indicated recent arrival in the area (resident for less than 2 years). Among those tested during intervention 2, 67% (587/879) were not registered during intervention 1. These indicate that conducting TTd twice led to increase in coverage as those not tested during intervention 1 were able to be tested in intervention 2. Eligible people who did not come for the skin snip test either refuse or were absent. In both intervention rounds, considered separately, almost 54% representing 643 in round 1 and 879 participants in round 2 participated in the skin snip test (Fig 2). The infection rates were 24% (161) for intervention 1 and 29% (251) for intervention 2. Treatment start rates were 71% (114) for intervention 1 and 83% (208) for intervention 2. The treatment completion rates were similar in the two rounds, being 93% (106) for intervention 1 and 98% (203) for intervention 2. These differences were not statistically significant.

A multiple logistic regression (MLR) was conducted including factors that appeared to influence participation in skin snip test, infection, and treatment (Table 2). During intervention 1, the likelihood of people participating in the skin snip test in Njinja/Njingouet was lower when compared to those in Makouopsap (OR = 0.03; p = 0.001; LR $\chi$2 = 79.53, p<0.001). Among the tested, the likelihood of infection was greater: among those in Makakoun compared to Makouopsap (OR = 2.5; p<0.001); among people aged 16–25 (OR = 2.2, p = 0.007) and 26–40 (OR = 2.5, p = 0.002) compared to the 9–15 years aged group; and among males (OR = 1.7, p = 0.004) compared to the female group (LR $\chi$2 = 41.82; p<0.00). During intervention 2, the likelihood of people participating in the skin snip test was lower: in

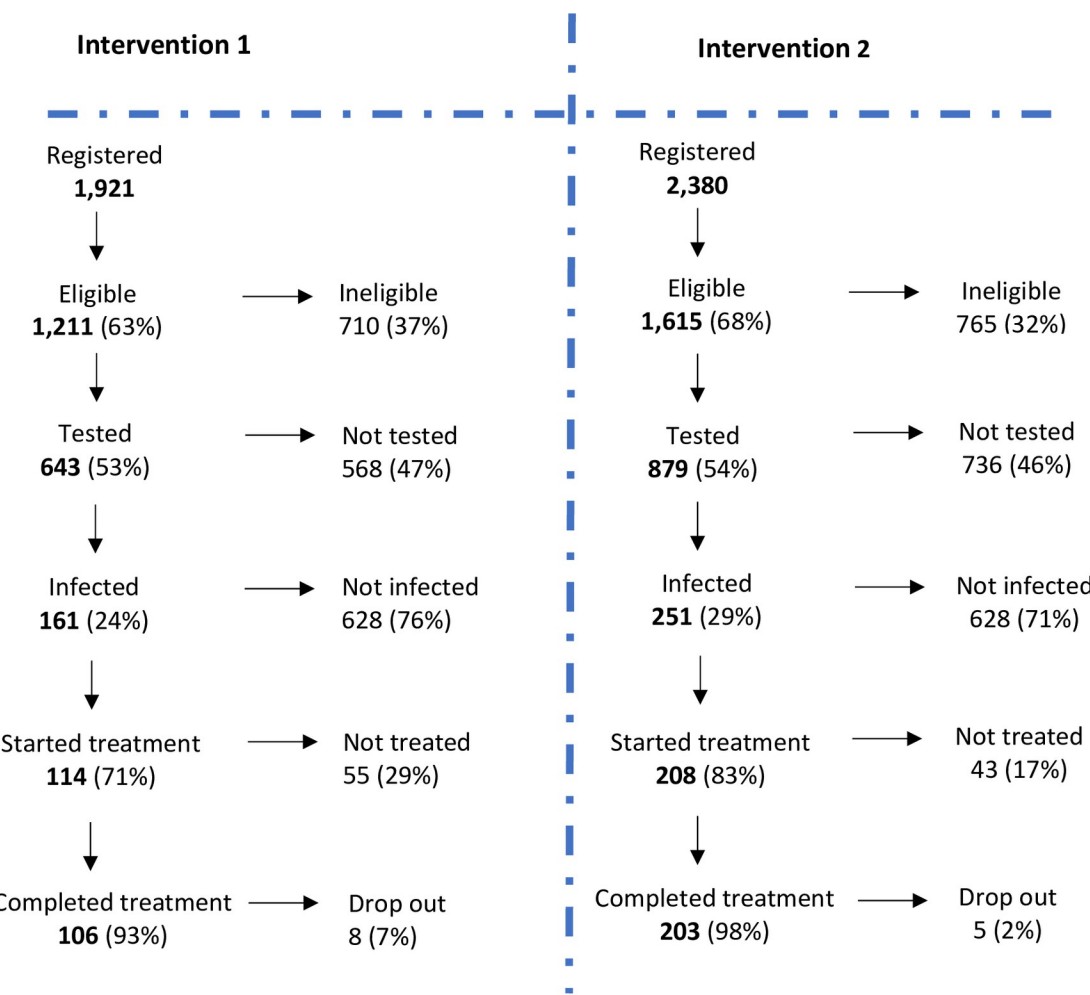

**Fig 2. Flow chart from registration to treatment for interventions 1 and 2.** Eligible criteria for skin snip test: >8 years, not pregnant, not breastfeeding, not severely ill; people not tested did not attend testing site, were absence or refused; drop out refers to those who started the treatment and did not complete; During intervention 1, 55 infected individuals did not start treatment among which 19 were underage (<9 years), 12 moved away; during intervention 2, 22 pregnant/breastfeeding women and 13 underage <9 years were excluded from treatment; 8 moved away after testing; during intervention 1; 1 person stop treatment because of side effects, 7 individuals moved away. During intervention 2, 5 persons moved away after starting treatment.

Makakoun (OR = 0.9; p<0.001), compared to Makouopsap; and for semi-nomads (OR = 0.4, p<0.001) compared to Bamoun (settled). However, the likelihood of participation was higher for those aged 26–40 (OR = 1.5, p = 0.024) and > 40 years (OR 2.3, p<0.001) compared to those aged 9–15 (LR χ2 = 106.62, p< 0.001). For infection rate, those in Makakoun likelihood of infection was lower (OR = 0.6, p = 0.014) compared to Makouopsap. Considering the main sub-populations, the likelihood of infection was lower among the semi-nomads (OR = 0.1, p<0.001) compared to Bamoun (settled). However, the likelihood of infection was higher for those aged 16–25 (OR = 2.3, p = 0.006), 26–40 (OR = 1.8, p = 0.038) and > 40 years (OR = 3.0, p<0.001) compared to those aged 9–15 years; as well as higher for males (OR = 1.6, p = 0.020) compared to females (LR χ2 = 82.36, p< 0.001) (Table 3).

In the FGD and KII qualitative assessment, participants mentioned they were motivated to participate in the skin snip test for several reasons including the desire to: know their infection

**Table 2. Intervention 1 and 2 screen and treatment participation levels.**

| Intervention 1 | | | | | | | | |
|---|---|---|---|---|---|---|---|---|
| Categories (data from participants aged ≥9 yrs) | Registered | Test participation | | Infection (positivity) | | Treatment start | | Course finalisation |
| Sub-categories | n (%) | n (%) | Odds[1] (P) | n (%) | Odds[2] (P) | n (%) | Odds[3] (P) | n (%) |
| **Overall** | | 1,211 | 643 (53) | | 161(24) | | 114 (71) | | 106 (93) |
| **Sex (1,211)** Females | 610 (50) | 317 (52) | ref | 66 (21) | **ref** | 37 (56) | **ref** | 34 (92) |
| Males | 601 (50) | 326 (54) | 1.1 (0.428) | 95 (29) | **1.6 (0.007)** | 77 (80) | **3.4 (0.001)** | 72 (94) |
| **Age groups (1,211)** 9–15 yrs | 291 (24) | 147 (51) | ref | 25 (17) | ref | 19 (76) | ref | 17 (89) |
| 16–25 yrs | 329 (27) | 155 (47) | 0.9 (0.398) | 41 (26) | **1.8 (0.036)** | 19 (46) | **0.3 (0.021)** | 17 (89) |
| 26–40 yrs | 308 (25) | 173 (56) | 1.3 (0.166) | 52 (30) | **2.2 (0.005)** | 38 (73) | 0.9 (0.784) | 36 (95) |
| >40 yrs | 283 (23) | 168 (59) | **1.4 (0.033)** | 43 (26) | 1.7 (0.051) | 38 (88) | 2.4 (0.190) | 36 (95) |
| **Communities (1,211)** Makouopsap | 427 (35) | 262 (60) | ref | 45 (18) | ref | 30 (67) | ref | 27 (90) |
| Makakoun | 504 (42) | 310 (61) | 1.0 (0.823) | 103(34) | **2.3 (<0.001)** | 74 (72) | 1.3 (0.527) | 69 (93) |
| Njinja/Njingouet | 280 (23) | 102 (35) | **0.3 (<0.001)** | 13 (14) | 0.8 (0.564) | 10 (77) | 1.7 (0.484) | 10 (100) |
| **Community stay (1,211)** < 2 yrs | 114 (9) | 50 (44) | ref | 3 (6) | ref | 1 (33) | ref | 1 (100) |
| 2–5 yrs | 149 (12) | 96 (64) | **2.3 (0.001)** | 14 (15) | 2.9 (0.106) | 9 (64) | 3.6 (0.341) | 9 (100) |
| 6–10 yrs | 216 (18) | 120 (56) | **1.6 (0.044)** | 27 (23) | **4.8 (0.014)** | 15 (56) | 2.5 (0.476) | 14 (93) |
| > 10 yrs | 763 (63) | 377 (52) | 1.4 (0.130) | 117 (31) | **7.1 (0.001)** | 89 (76) | 6.4 (0.571) | 82 (92) |
| **Intervention 2** | | | | | | | | |
| Categories (data from participants aged ≥9 yrs) | Registered | Test participation | | Infection (positivity) | | Treatment start | | Course finalisation |
| Sub-categories | n (%) | n (%) | Odds (P) | n (%) | Odds (P) | n (%) | Odds (P) | n (%) |
| **Overall** | | 1,615 | 879 (54) | | 251 (29) | | 208 (83) | | 203 (98) |
| **Sex (1,615)** Females | 795 (49) | 410 (52) | ref | 96 (24) | ref | 67 (70) | ref | 63 (94) |
| Males | 820 (51) | 469 (59) | **1.3 (0.023)** | 155 (33) | **1.6 (0.002)** | 141 (91) | **4.4 (<0.001)** | 140 (99) |
| **Age group (1,615)** 9-15yrs | 432 (27) | 213 (49) | ref | 30 (14) | ref | 24 (80) | ref | 21 (88) |
| 16-25yrs | 388 (24) | 177 (46) | 0.9 (0.291) | 49 (27) | **2.4 (0.001)** | 33 (67) | 0.5 (0.227) | 33 (100) |
| 26-40yrs | 453 (28) | 258 (57) | **1.4 (0.023)** | 82 (32) | **2.9 (<0.001)** | 68 (83) | 1.2 (0.720) | 68 (100) |
| >40yrs | 342 (21) | 231 (68) | **2.1 (<0.001)** | 90 (39) | **3.9 (<0.001)** | 83 (92) | 3.0 (0.071) | 81 (98) |
| **Community (1,615)** Makouopsap | 535 (33) | 310 (58) | ref | 108 (33) | ref | 92 (85) | ref | 88 (96) |
| Makakoun | 678 (42) | 332 (49) | **0.7 (0.002)** | 106 (32) | 0.9 (0.449) | 85 (80) | 0.7 (0.351) | 84 (99) |
| Njinja/Njingouet | 402 (25) | 237 (59) | 1.0 (0.756) | 37 (16) | **0.3 (<0.001)** | 31 (84) | 0.9 (0.838) | 31 (100) |
| **Community stay (1,615)** < 2 yrs | 65 (4) | 41 (63) | ref | 4 (10) | ref | 4 (100) | – | 4 (100) |
| 2–5 yrs | 417 (26) | 187 (45) | **0.5 (0.007)** | 33 (18) | 2.0 (0.222) | 29 (88) | 1.4 (0.514) | 28 (97) |
| 6–10 yrs | 231 (14) | 121 (52) | 0.7 (0.127) | 34 (28) | **3.6 (0.023)** | 25 (74) | 0.6 (0.179) | 24 (96) |
| > 10 yrs | 904 (56) | 530 (59) | 0.8 (0.495) | 180 (34) | **4.8 (0.003)** | 150 (83) | 1.0 (—) | 147 (98) |
| **Tribes (1025)** Bamoun | 494 (48) | 298 (60) | ref | 119 (40) | ref | 96 (81) | ref | 93 (97) |
| Semi-nomads | 212 (21) | 102 (48) | **0.6 (0.003)** | 7 (7) | **0.1 (<0.001)** | 7 (100) | – | 6 (85) |
| Others | 319 (31) | 174 (55) | 0.8 (0.103) | 74 (43) | 1.1 (0.531) | 63 (85) | 1.4 (0.430) | 63 (100) |

1 = Odds/probability of participating in the skin snip test; 2 = Odds/probability of having infection; 3 = odds/probability of starting treatment. **Subcategories highlighted have crude OR significantly different from that of the reference.** During each intervention rounds, census was conducted. Only number of aged-eligible (≥9 years) are noted in this table as registered. Individuals were invited for skin snip test (Eligible criteria for skin snip test: >8 years, not pregnant, not breastfeeding, not severely ill). Those tested are noted as test participation. If infected (presence of onchocercal microfilaria (mf)), the infected individuals are noted as infection (infectivity)). The infected and eligible individuals were offered 35-day course of doxycycline and the number of people starting (treatment start) and completing treatment (course finalisation) noted. Absolute numbers and percentages for sex, age group, community (of resident), duration of stay in the community (community stay) and tribes (only for intervention 2) categories are reported alongside crude ratios (OR) and its p-values (p) with respect to a reference (ref) or based subgroup within the category.

**Table 3. Intervention 1 and 2 test, infection and treatment rates comparison within categories.**

| Outcome | Categories | Subcategories | Intervention 1 | | | Intervention 2 | | |
|---|---|---|---|---|---|---|---|---|
| | | | Adjusted OR[1] | p-value | Fitness of Regression Model | Adjusted OR[2] | p-value | Fitness of Regression Model |
| Test rate[1] | Community | Makouopsap | Ref | – | LR χ2 = 79.53; p<0.001 | Ref | – | LR χ2 = 106.62, p< 0.001 |
| | | Makakoun | 1.0 | 0.901 | | **0.9** | **<0.001** | |
| | | Njinja/Njingouet | **0.3** | **0.001** | | 1.89 | 0.086 | |
| | Age group | 9-15yrs | Ref | – | | Ref | – | |
| | | 16-25yrs | 0.9 | 0.348 | | 1.1 | 0.728 | |
| | | 26-40yrs | 1.2 | 0.207 | | **1.5** | **0.024** | |
| | | >40yrs | 1.4 | 0.062 | | **2.3** | **<0.001** | |
| | Sex | Females | Ref | – | | Ref | – | |
| | | Males | 1.0 | 0.934 | | 1.1 | 0.394 | |
| | Tribes[3] | Bamoun | – | – | | Ref | – | |
| | | Semi-nomads (Bororo) | – | – | | **0.4** | **<0.001** | |
| | | Others | – | – | | 0.9 | 0.576 | |
| Infection rate[2] | Community | Makouopsap | Ref | – | LR χ2 = 41.82; p<0.001 | Ref | – | LR χ2 = 82.36, p< 0.001 |
| | | Makakoun | **2.5** | **<0.001** | | **0.6** | **0.014** | |
| | | Njinja/Njingouet | 0.9 | 0.662 | | 0.5 | <0.180 | |
| | Age group | 9-15yrs | Ref | – | | Ref | – | |
| | | 16-25yrs | **2.2** | **0.007** | | **2.3** | **0.006** | |
| | | 26-40yrs | **2.5** | **0.002** | | **1.8** | **0.038** | |
| | | >40yrs | 1.7 | 0.06 | | **3.0** | **<0.001** | |
| | Sex | Females | Ref | – | | Ref | – | |
| | | Males | **1.7** | **0.004** | | **1.6** | **0.020** | |
| | Tribes | Bamoun | – | – | | Ref | – | |
| | | Semi-nomads (Bororo) | – | – | | **0.1** | **<0.001** | |
| | | Others | – | – | | 1.1 | 0.556 | |
| Treatment rate[3] | Community | Makouopsap | Ref | – | LR χ2 = 31.39; p<0.001 | Ref | – | LR χ2 = 34.57, p< 0.001 |
| | | Makakoun | 0.9 | 0.779 | | 0.6 | 0.257 | |
| | | Njinja/Njingouet | 2 | 0.388 | | 0.7 | 0.464 | |
| | Age group | 9-15yrs | Ref | – | | Ref | – | |
| | | 16-25yrs | 0.3 | 0.039 | | 0.6 | 0.415 | |
| | | 26-40yrs | 1.1 | 0.899 | | 1.3 | 0.689 | |
| | | >40yrs | 3.1 | 0.105 | | **4.1** | **0.029** | |
| | Sex | Females | Ref | – | | Ref | – | |
| | | Males | **3.7** | **0.001** | | **5.1** | **<0.001** | |

1: OR stand for odd ratio or probability of participating in skin snip test during round 1; 2: OR stands for odd ratio or probability of participation in the skin snip test during round 2; 3: variable tribe was included only in round 2 following field observation that there were minority semi-nomadic population present, and their inclusion appear challenging. During each intervention round, census and skin snip test (eligibility criteria for skin snip test: ≥8 years, not pregnant, not breastfeeding, not severely ill) were conducted. If infected (presence of onchocercal microfilaria (mf) in the skin snip), they were offered 35-day course of doxycycline and the number of individuals starting (treatment start) and completing treatment (course finalisation) noted. Adjusted OR and LR chi-square test (χ2) (indicating fitness of regression model) are reported alongside their p-values for both interventions. The highlighted categories indicate those having ORs which are significantly different from the reference (ref) categories

status; have better health; and protect others. Some respondents detailed that participation showed a sense of responsibility, solidarity, and care for community wellbeing.

*"That (test) permitted my illness to be revealed. Therefore, for me it was good–especially that I did not have any problem following that. And everyone was happy with the result given that everyone knows that if they are positive, they will be offered treatment"* (Settled community-based participant).

Forty six percent (46%) of people in each round did not participate in the skin snip test during intervention 1. The levels of non-participation were not uniform with regards to age and community categories. In a multiple logistic regression (Table 3) the likelihood of participating in the test were lower in community of Njinjia/Njingouet (OR 0.3; p = 0.001) in intervention 1, and in communities of Makakoun (OR 0.9; p = 0.001) during intervention 2 when compared to Makouopsap community (Table 2). The likelihood of participation, compared to 9–15 years age group, were significantly higher among those in the 26–40 (OR 1.5; p = 0.024) and >40 (OR 2.3; p<0.001) age groups in intervention 2.

Qualitative assessment through FGD and KII of programme staff and community members, reveal reasons for non-participation in skin snip test. Some youth of the settled communities felt they were not sick nor at risk and saw no reason to bear the wound, pain, or any unforeseen negative impact of the test. Reportedly, their trust and confidentiality were being influenced by speculation that testing apparatus will "shout out" all the sickness in their body audibly in public. Participants mentioned that collected skin snips are intended for malicious use to obtain mystical power. Report of previous tests with no results provided was said to be demotivating. The lack of results from tests was also cited as supporting evidence for the belief that skin samples were used to obtain mystical power. Furthermore, some participants doubted the test results; questioning that they experience symptoms of onchocerciasis as detailed in sensitisation yet have negative results.

*"I refused. . .. Why are they cutting again this time around? I did not take treatment [last time]. They told me I have no illness in me. Meanwhile I know that I have illness such as eye pain, skin rashes and filaria [onchocerciasis]. For someone like me–look at my eyes, I also have skin rashes but to my greatest surprise they said I was negative. It surprised me"* [semi-nomadic chief].

To address concerns raised, adjustments were made for intervention 2 including reinforced social mobilization and sensitization using a mobile musical van to make announcements about benefits of the intervention and invite people to participate. In addition, microscopic examination of samples was done at the sampling point with immediate return of result and inviting positive participants to view live microfilariae through microscope. This led to additional participation in intervention 2 evident by quantitative findings that 15% (83) of those declining to participate in intervention 1 participated in intervention 2. In addition, 64% (566) of those participating in the test were people who had not been registered in intervention 1 round despite indicating they had been resident in the study area at the time (resident for more than 2 years). One hundred and sixty-nine (169) people participated in both skin snip tests. Participants in the FGD and KII equally mentioned the increased in participation.

*"When people were seeing in the microscope, they were shocked to know that microbe lives in them. These made adhesion [to testing by the population] total! People like concrete proof. Because . . .. you say take this medicine and they reply, "prove to me I am sick". Them seeing the microbe under the microscope–no question anymore"* (District Medical Officer).

During intervention 1, a minority semi-nomadic subgroup with a considerable population size was noticed in the study area. They appeared not to have been equitably reached during house-to-house census or TTd. In intervention 2, efforts were made to reach these semi-nomads. However, the MLR indicated semi-nomads were still less likely to participate when compared to the settled (Bamoun tribe) population (OR 0.4; p<0.001, Table 3). FGD and KII assessment revealed semi-nomadic specific barriers to participation: their remote, dispersed and small settlements made engagement difficult; their high mobility (seasonal and daily) lead to absence during drug distribution; the lack of inclusion of semi-nomads as Community-Directed Distributors (CDDs) in routine MDA reinforced low acceptance through dis-trust; linguistic and cultural differences between CDDs and semi-nomads also contributed; limited knowledge about semi-nomads leading to their inadequate consideration during MDA programme planning; low awareness of onchocerciasis and MDA and low education (literacy) levels among semi-nomadic community members were also mentioned as barriers.

*"The difficulty is more communicational. When we arrive at the health centre, we are unable to make ourselves understood by the nurses and we are also unable to understand what they are saying to us. This affects us a lot as very often the disease that we consult for is not what is being treated due to faulty comprehension between us and the nurses"* [Female semi-nomad].

*"I face some difficulties, I gave drug to one person [a semi-nomad], saying take one for [every] one day, instead he was taking two per day as he wanted the sick[ness] to finish quickly. He did so for 5 days. I was afraid. The difficulty now is to go and reach their place. There is problem: water, bush, all those things. You know, there was a time all road to their place were muddy and flooded. At times, the motorbike will be stuck, and you will need to trek. . .."* [male settled community based CDD].

## Treatment participation

Treatment participation levels in the two intervention rounds are found in Table 2. During intervention 1, males where more likely to start treatment than the females (OR = 3.7, p = 0.001, LR $\chi 2$ = 31.39; p<0.001). For intervention 2, the likelihood of starting treatment was again higher for males (OR = 5.1, p<0.001) compared to the females and higher among those aged >40 compared to the 9–15 aged group (OR = 4.1, p = 0.029; (LR $\chi 2$ = 34.57, p< 0.001).

In intervention 1, a delay between testing and treating people contributed to a loss to follow-up as only 71% of infected participants started treatment. The census, testing and treating were streamlined during the second round, effectively reducing attrition. In intervention 2, nine months following end of intervention 1 treatment, some participants found positive self-reported they had positive result and received treatment during intervention 1 (34 individuals; however, 7 only were found in matched record). In response, a quality assurance of the doxycycline including verification of dosage of active ingredient was performed, which was passed, ahead of the second round.

From qualitative findings, participants were appreciative of the meal support, reporting that the treatment arrangement was suitable and aided their daily activities.

*"Every morning, we took café or "pap" [a type of corn-based semi-liquid food] then we took the medication, and everyone was good for the rest of the day. We were not really disturbed by the time of the treatment as it never prevented one from going about daily activities especially as the treatment points were not far from our houses. None of us were really making big efforts to go to the treatment point".* [settled community-based participants].

Furthermore, treatment participation was frequently described as a positive experience with perceived improvement to health.

*"I took [doxycycline]. I even felt certain internal bumps in the body; but now I see external, I feel it's going out, pushing and going out.... There is no problem. But we feel that it* [doxycycline] *is pushing this disease in the body out. And I know when it's out, any moment it's going to end. When it is hiding, it will disturb years and years in the body."* [Settled community-based male]

## Relating ivermectin and doxycycline treatments

Some participants compared ivermectin and doxycycline treatments with some reporting ivermectin as better than doxycycline for alleviating symptoms; meanwhile others mentioned doxycycline is better.

*"It [ivermectin] has helped people a lot. At first, I was seeing small bumps on my body, as I consume ivermectin, I realised that the bumps have completely disappeared. Ivermectin has helped many people. So, if you want to help us, bring more ivermectin because I don't really understand the point of doxycycline"* [settled community beneficiary].

*"...they noted changes, for example, we have someone here who constantly complaint of body itches, when he took doxycycline, it disappeared. This person was taking Ivermectin previously"* [CDD].

Programme staff expressed concern about timing between of ivermectin and TTd, citing a lack of clarity on how ivermectin will impact TTd. This resulted to testing in intervention 1 being less than three months since CDTi, suspected to have contributed to low detected infection level during intervention 1 compared to baseline (Intervention 1: 24%, n = 643, Table 2 vs baseline: 36%, n = 557) [16, 18]. This prompted a catch-up round (intervention 2) where everyone was offered testing and results revealed a higher infection level than in round 1 (round 2 being 29%, n = 879, Table 2).

## Treatment burden on the community-directed drug distributors

CDDs reported the treatment was a burden as the treatment process took their whole day over a 35-day duration, restricting their daily activities, with insufficient financial compensation.

*"The difficulty of the CDD is that the 35 days is not easy. You are obliged to always remain on the spot..... Because the small thing [money] that they give us, does not meant that it can compensate our day of work. We can say the 35 days is sacrifice as the CDD is incapable of carrying out their own activities–go do other things, travel etc. You are obliged to be around during the 35 days and after the 35 days if there is catch-up."* [Female settled community based CDD].

During treatment, settled community based CDD struggled to reach semi-nomadic camps daily. In response, doxycycline was delivered weekly to semi-nomadic communities. Drug packets were requested for verification during each visit to ensure adherence. Although the CDD appreciated these adaptations, they raised cases where patients misunderstood dosing, which was not promptly resolved due to the weekly visits.

*"I face some difficulties, I gave drug to one person [a semi-nomad], saying take one for one day, instead he was taking two per day as he wanted the sick[ness] to finish quickly. He did so*

*for 5 days. I was afraid. The difficulty now is to go and reach their place. There is problem: water, bush, all those things. You know, there was a time all road to their place were muddy and flooded. At times, the motorbike will be stuck, and you will need to trek. . .. The seven days was a good idea, but you are not sure if s/he has taken for the seven days. . .. When you give for a week and you go, upon returning, you still have two tablets left when the week has finished"* [male CDD].

## Side effects of doxycycline treatment

Records of side-effects in the registers indicate that 15% (13) of participants reported side-effects during intervention 1. Reports included headache, dizziness, stomach-ache, but none of them stopped treatment or sought medical attention in the hospital. During intervention 2, 25% (55) of participants reported side effects which did not necessitate stopping treatment or consultation. Similar side effects were mentioned in the qualitative assessment and there was equally no mention of consulting in health facilities.

*"There were no side effects. Nobody came for that; nobody. I think as they ate before swallowing the drug, that is the reason; no side effect after 35 days of treatment with doxycycline. That is strange!"* [District Medical Officer].

## Beneficiary feedback from the field for improvement

Improvement was proposed around gender inclusion, by selecting more female CDDs. In addition, female associations should be prioritized for sensitisation during intervention. Participants reported absenteeism as reason for some participation gaps, suggesting more time be allowed to offer more people opportunity to participate. This extended timeframe would benefit semi-nomadic communities, who are often absent because of their daily and seasonal mobility. Benefits will also apply to the settled community members who miss testing due to absence.

*"Like those who came to claim the ivermectin from me [afterward], I could no longer [give]; I could still give with what permission? I could not. Because when they [semi-nomads] arrive, they are informed by the old ones; [and] they come to us to ask [theirs]. Well, then there is nothing we can do. . . We do not have permission [to give because distribution period is over]"* [Female settled community based CDD].

Some CDDs and semi-nomads reflected that the programmatic set up or delivery model were sub-optimal for semi-nomadic populations. Semi-nomads requested that the sensitisation contents, support, channels and language are adapted to include their needs, ideally, they should be directly included in planning and implementation.

*"For us Bororo [Fulani semi nomads], if you assign us this task, to one or more Bororos, it will be very easy to mobilize a lot more people. They will be able to say to theirs, in a convincing manner, what you want to pass on as message. Even those absent will be well informed, given that we know each other in detail. There are some of us that you cannot see but we know where and when to meet them and above all, how to speak to them to participate. Some of us were present at the Centre during the test but did not do [participate] because there was nobody to better explain to them the reasons for the test, the procedure, and the risk"* [Female semi-nomadic participants].

Participants believed that more systematic local engagement through local chief, camp leaders and inclusion of semi-nomads as CDDs will improve participation. Outreach to semi-nomadic camps testing, moving from camp to camp, staying on site until absent residents return home, if necessary, was suggested by semi-nomadic chiefs as a way of improving inclusion.

## Discussion

Skin snip test participation rate was 54% during each round increasing to a minimum of 57% (based on matched record) and a maximum of 83% (based on self-reported participation) when both rounds are considered. This participation level is close to 60.5% obtain in Mbanga and Melong Health Districts of Cameroon where doxycycline treatments were offered without prior testing [15]. An approach we did not attempt due to required cost, long duration of treatment and logistics demands. An additional 46% of individuals was censused during intervention 2. This highlights the challenges of understanding the true census/denominator of the target population, especially as a significant semi-nomadic population was present in the area, whom were not systematically engaged during intervention 1. However, we believe the 46% is inaccurately excessive due to a high level of mismatch between registers of intervention 1 and 2, which was conducted manually based on location, names, age, and sex. Such mismatching led to an underestimate in cumulative population test participation over 2 rounds by inflating the denominator population. Therefore, the 57% test participation rate is considered a conservative minimum. The remaining 43% is significant especially if prevalent of infection in them is higher than in those who participated—something possible as those not participating may have not been participating in usual MDA given that the same platform is used.

Previous onchocerciasis research activities that took skin snips from people in the area [16, 17] may have biased the population's self-reported past participation. Recall bias was minimised by asking participants to specify the year that tests were conducted; only responses citing the relevant year were included in analysis. Thus, participation over the two rounds is near 83%. This high participation was due to the repetition of the test, often people who refused in round 1 gained trust from talking with peers who had tested, and then accepted to participate during round 2 that also benefited from reinforced sensitization. There were also newcomers arriving into the area after round 1 and participating in round 2. Qualitative data revealed concerns surrounding health and disease status as common motivator for test participation. In connection, TTd resistance from younger people stemmed from them feeling healthy and not needing treatment or an invasive and painful test. Quantitative findings backed up the qualitative findings, revealing older age group testing participation was more likely than the younger population (OR >1; p<0.05). This study found high infection (OR >1 and p<0.05) among older age categories, who are therefore more likely to experience onchocerciasis-related signs and symptoms driving increased test acceptance [24].

Low participation amongst the semi-nomadic population became apparent during the implementation. Despite adjustments to include semi-nomads, the relative participation level was 40% lower compared to the settled population, this highlights the severity of equity issues. Qualitative findings confirmed and explored reasons for inequity, revealing mistrust, ill-adapted sensitisation, programme reach issues linked to use of CDDs from settled community, their frequent mobility and their remotely dispersed small settlements. Inequity factors are compounded by inherent programmatic issues, not adequately incorporating semi-nomadic challenges into programme strategies. The semi-nomads' mistrust of the settled population is rooted in conflict and discrimination frequently stemming from land access conflict. Not reaching these semi-nomads threatens the impact of the intervention and its sustainability.

Systematic programme planning including semi-nomads, reinforced engagement, adaptations of sensitisation (channels, language, and content), inclusion of semi-nomadic CDDs as well as outreach to their settlements are strategies to consider.

There was inter-community variation in participation in skin snip test. During intervention 1 people in Njinja/Njingouet participated less in the test than their Makouopsap counterparts. This pattern disappeared during intervention 2 and instead participation rate was lower for the Makakoun community. The lower participation in Njinja/Njingouet may have been due the semi-nomads refusing in the first instance as reported during FGD and KII that nomads in Njinja/Njingouet only recently arrived and more conservative than those in the other communities. In addition, awareness of onchocerciasis including skin snip test was lower among them. Similarly, in Makakoun, residents are mainly non-ethnic Bamoun. Youths in this community expressed mistrust and refusal of the skin snip test during intervention 2 which may have impacted skin snip participation rate. The reinforced sensitisation during intervention 2 may have reduced inter-community disparity with the semi-nomads in Njinja/Njingouet. But this became prominent in Makakoun which may have been due to lack of sensitization targeting the youth.

Concerns surrounding confidentiality and trust of the test were reported among the wider population. Qualitative assessment reveal beneficiaries' perspectives such as getting a negative result when symptomatic, impacted their acceptability of the programme. Mistrust appeared to be the underlining explainer of variation in participation within community. This mismatch has been partially due to testing occurring within less than three months following administration of the microfilaricide ivermectin, which renders infection undetectable by skin snip. However, reduction in skin mf due to ivermectin would have reduced the symptoms given that mf are responsible for most onchocerciasis-related symptoms [25, 26]. It is possible that some of the symptoms may not be onchocerciasis related, especially as control of onchocerciasis-related morbidity has been achieved in many places [3]. Alternatively, symptoms could be onchocerciasis symptoms remembered from before ivermectin administration. Limitations surrounding sensitivity of skin snip microscopy test [27] means that continued presence of mf undetectable by skin snip microscopic test cannot be ruled out with 100% certainty.

We reiterate the need for a less invasive and more sensitive test to detect active onchocerciasis infection. Such diagnostic is needed when high mf intensity would be suppressed in the skin and symptoms are unlikely to be encountered but transmission ongoing with vector surviving even better when low load of mf is ingested [28]. Such a test is also necessary for point of care management of onchocerciasis when MDA will stop. In instances of treatment naïve individuals or settings (such as hypo-endemic places—not previously considered for treatment), use of the more acceptable Ov-16 Rapid Diagnostic Test (RDT) should be considered [29]. Sensitization materials should be reviewed, considering current epidemiology of onchocerciasis alongside beneficiary perspective. In line with these, a holistic approach for skin health should be considered where possible.

Previous experience also influenced acceptability of TTd. People complained previous testing that they never received results or treatment from. This could have been related to our study, in delay in intervention 1 between testing, results, and treatment which led to a low (71%) treatment start level. This was due to delay obtaining necessary logistics (punches for skin snip and doxycycline) compounded by inadequate coordination and planning among implementation partners. Equally, before this TTd intervention, the same area has been surveyed using skin snip [16, 17] which could be conflated in people's memories due to similarity. Further compounding, instances of un-returned results led people to hold negative assumptions about the test, including skin being used for magical rituals. This contributed to refusal rates, especially during intervention round 1. Improvements made in the second round

through reinforced sensitisation, immediate result return, demonstration of live mf and immediate start of treatment improved the situation.

Once tested, doxycycline treatment was well accepted, reflected in high treatment start (83%) and completion (98%) levels of intervention 2. The lower start rate (71%) in intervention 1 resulted from delays in starting treatment and were thought to be due to semi-nomads and farmers relocating. During treatment, side effects were low and mild throughout the study as participants received a meal with the doxycycline tablet. Along with the intended reduction in side effects, meal support enhanced acceptability of treatment by acting as an incentive to attend for medicines. Eating and taking treatment in the morning fit well with beneficiaries' activities and even enhanced their daily activity. Contrarily, CDDs stated that treatment put more burden on them as they often forwent their daily activities. Treatment options such as weekly approach for the semi-nomadic population or increase CDD compensation package should be considered.

TTd could influence the acceptability of community-directed treatment with ivermectin (CDTi) as there were concerns that the largely popular CDTi would be replaced by TTd. Some respondents mentioned that ivermectin is better than TTd, driven by beneficiary perspectives of health improvement after taking ivermectin. This view corresponded to ivermectin's rapid reduction of skin mf which alleviates symptoms [30]. Some people stated that doxycycline did not improve their health. This aligned with findings that some people tested and treated in intervention 1 were still found infected in round 2. This may be due to doxycycline's longer time to alleviate symptoms by impacting mf production, since mf can stay in the skin for up to 12 months [9, 13] and intervention 2 occurred before the next MDA. This supports the need of combining TTd with ivermectin [13] where loa loa is not a risk factor. Implementing CDTi immediately after skin snip collection followed by doxycycline treatment to people with detected mf, would give beneficiaries immediate and sustained symptoms clearance [31]. It will also be more impactful by breaking the transmission cycle quicker as evident in the impact evaluation of these interventions and elsewhere [18, 32]. There are some evidence that even though mf are still present for >12 months after doxycycline treatment, their ability to be transmitted is hindered [8, 14]. In addition, combining ivermectin and TTd will be cost effective by combining logistics, ensuring normal CDTi continues seamlessly during and after TTd intervention. Ivermectin drug fatigue and changing community priorities are becoming apparent in many places in sub-Saharan Africa [33–35] and any strategy such as TTd that build on the CDTi platform are likely to inherit challenges. There is need to fully understand these challenges even if medicine is changed. However, as a cure, it will not be necessary to continue implementation of TTd for more than a couple of rounds making it less susceptible to some ivermectin related challenges such as drug fatigue.

Community-led TTd is feasible and require one round of treatment as it is curative, something that CDTi will achieve in at least 10 years. The side effect was effectively monitored by CDD and reported to be 15% among the participants. This is higher for reported effect of ivermectin at standard dose for onchocerciasis (1.5%) [36]. However, the reported side effects were short-lived, and did not require attending health facilities. Considering single round of CDTi and TTd, TTd is far more demanding in terms of logistic, cost, training, expertise, and duration. However, taking into consideration the required rounds to cure (disappearance of adult worm), CDTi required much longer duration–at least 10 years which commutatively will surpass the burden of TTd in one or even two rounds. During this time, new challenges can set in making the CDTi more challenging and costly to implement and even prolonging the require duration to achieve elimination. Such events include immigration such as the semi-nomads witnessed in this Massangam and the development of new man-made breeding sites due to building dams and bridges. Equally, prolong administration of a drug make it prune to

resistant of the agent and fatigue for those taking it—something already being witness with ivermectin as mentioned earlier.

In terms of community acceptability, a new and invasive test achieving 54% of eligible population is significant and if it became a programmatic strategy, coverage can be higher. But TTd should aim at two rounds instead of multiple rounds as CDTi. The second round is a mop, and the combined coverage will be high as seen in this study with over 80% coverage which is comparable to the recommended minimum coverage for CDTi. Skin snip test are routinely done in district hospitals in endemic areas. These hospitals' capacity can be reinforced to improve skin snip test and they can conduct TTd as a community-led outreach strategy. Skin snip test can be waived in places of very high prevalence of onchocerciasis that may warrant blanket doxycycline treatment as implemented by Wanji and colleagues [15] in Mbanga and Melong Health Districts of Cameroon. In ivermectin naïve places, use of less invasive rapid diagnostic test such as the OV16 RDT should be considered. However, it has been shown that parasitological (skin biopsy) and immunological (OV16 antigen) parameters are in moderate agreement [37]. Therefore, new diagnostics being develop such as multiplex, loop mediated isothermal amplification (LAMP) and others are of interest [38–40]. In any acceptability of doxycycline treatment will be much higher in a single round than when skin snip is involved as it is the main determinant of TTd acceptability. Whereas CDTi covers 5–8 years old individuals, doxycycline excludes this population. This adds to the need to implement ivermectin alongside TTd for this population in areas where loa loa co-endemicity is not a risk. There is a need to conduct more research including modelling of doxycycline treatment to guide programme to take strategic option as to where, how and for how long to implement doxycycline treatment.

We conclude that TTd is acceptable and achieved a significant participation level (57%) for a curative intervention. Though lower than the 65% recommended ivermectin coverage [3, 41], it impact is felt in shorter term [16, 18]. However, we appreciate that the remaining 43% is still significant especially if distribution of mf infection is skewed toward these non-participants. It has led to a significant and rapid reduction in mf [16, 18]. In this paper, we have highlighted the value of TTd and its consideration in future endeavours, with the following recommendations.

Recommendation 1: Planning and preparation. Planning should begin by exploring the community to identify potential participation issues including systematic refusal, census underestimates, minority, and frequently excluded groups.

Recommendation 2: Improving test sensitivity and adopting a holistic approach. Research to find more sensitive tests should continue and be used when available. In addition, TTd approach should incorporate general skin consultation to address participant's self-diagnosis of onchocerciasis based on general symptoms of skin diseases but testing negative.

Recommendation 3: Combine CDTi and TTd, optimizing their timing. As doxycycline has little or no effect on mf already produced, given it together with ivermectin once will clear these mf which may become residual. This also ensure continuation of CDTi after TTd until elimination is ascertained. In addition, people 5–8 years old, ineligible to doxycycline, would continue to benefit from ivermectin treatment. Thus, while implementing TTd, maintain CDTi simultaneously, testing and offering ivermectin treatment the same day along with starting doxycycline treatment immediately once the result is available. A follow-up TTd round could be added.

Recommendation 4: Update and adapt sensitization. Review, update and adapt sensitization methods, channels and content considering participants' needs. Include showing participants microfilaria under the microscope and ensure all messages are relevant from beneficiary perspectives and relatable by different population subgroups.

Recommendation 5: CDD motivation and compensation. Method of medicine delivery should be considered for modification to reduce burden on volunteers dispensing the medicine. Alternatively, improve compensation for CDDs as TTd is more time demanding than CDTi.

## Supporting information

**S1 Text. Advocacy meeting report.**
(PDF)

**S2 Text. Round 1 field monitoring report.**
(PDF)

**S3 Text. Review and planning meeting report.**
(PDF)

**S4 Text. Key Informant Interview (KII) and Focus Group Discussion (FGD) guides.**
(PDF)

**S1 Table. Table showing timeline of implementation of Alternative Treatment Strategies package (ground larviciding, biannual ivermectin mass drug administration and test and treat with doxycycline.**
(PDF)

**S2 Table. Doxycycline treatment register.**
(PDF)

**S3 Table. Key Informant Interview (KII) and Focus Group Discussion (FGD) findings (qoutes).**
(XLSX)

## Acknowledgments

We remain indebted to NTD programme in Cameroon at National, Regional and District levels and the communities who supported all the activities. We are also thankful to the Centre for Research on Filariasis and other Tropical Diseases (CRFilMT) for performing sample collection and laboratory analysis. We thankfully recognise the generous contribution of the COUNTDOWN consortium/ Liverpool School of Tropical Medicine in the development of this protocol. The consortium provided insightful advice during the study's implementation. We appreciate Alexandre Chailloux for preparation of maps, and Susan D'Souza and Louise Hamill for their review of the manuscript.

## Author Contributions

**Conceptualization:** Benjamin Biholong, Samuel Wanji, Didier Bakajika, Joseph Oye, Elena Schmidt, Laura Senyonjo.

**Data curation:** Rogers Nditanchou, Ruth Dixon.

**Formal analysis:** Rogers Nditanchou, Ruth Dixon, Aude Wilhelm, Sapana Basnet.

**Funding acquisition:** Ruth Dixon, Didier Bakajika, Elena Schmidt, Laura Senyonjo.

**Investigation:** Rogers Nditanchou, Kareen Atekem, Serge Akongo, Franklin Ayisi, Philippe Nwane, Samuel Wanji, Joseph Kamgno, Daniel Boakye, Laura Senyonjo.

**Methodology:** Samuel Wanji, Didier Bakajika, Laura Senyonjo.

**Project administration:** Rogers Nditanchou, Ruth Dixon, Kareen Atekem, Serge Akongo, Benjamin Biholong, Joseph Oye, Laura Senyonjo.

**Resources:** Elena Schmidt.

**Supervision:** Kareen Atekem, Benjamin Biholong, Joseph Kamgno, Daniel Boakye, Laura Senyonjo.

**Writing – original draft:** Rogers Nditanchou, Laura Senyonjo.

**Writing – review & editing:** Rogers Nditanchou, Ruth Dixon, Kareen Atekem, Franklin Ayisi, Richard Selby, Didier Bakajika, Daniel Boakye, Elena Schmidt, Laura Senyonjo.

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
