## [Decision Letter · Decision Letter 0]

11 Oct 2022

Dear Dr. NDITANCHOU,

Thank you very much for submitting your manuscript "Acceptability and effectiveness of test and treat with doxycycline against Onchocerciasis in an area of persistent transmission in Massangam Health District, Cameroon" for consideration at PLOS Neglected Tropical Diseases. As with all papers reviewed by the journal, your manuscript was reviewed by members of the editorial board and by several independent reviewers. In light of the reviews (below this email), we would like to invite the resubmission of a significantly-revised version that takes into account the reviewers' comments. 

We cannot make any decision about publication until we have seen the revised manuscript and your response to the reviewers' comments. Your revised manuscript is also likely to be sent to reviewers for further evaluation.

Sincerely,

Maria Angeles Gómez-Morales, PhD

Academic Editor

Eva Clark

Section Editor

Reviewer's Responses to Questions

**Key Review Criteria Required for Acceptance?**

**Methods**

-Are the objectives of the study clearly articulated with a clear testable hypothesis stated?

-Is the study design appropriate to address the stated objectives?

-Is the population clearly described and appropriate for the hypothesis being tested?

-Is the sample size sufficient to ensure adequate power to address the hypothesis being tested?

-Were correct statistical analysis used to support conclusions?

-Are there concerns about ethical or regulatory requirements being met?

Reviewer #1: The authors should describe in detail what regression models were used, and what statistical analysis was conducted. 

The authors described that surveys and discussions took place through group discussions and Key Informant Interviews. The method used to include participants in these groups, the method used to conceive the content of these survey and the discussions should be described. Furthermore the whole content of these tools should be included in the manuscript (forms questions, answers of the participants, sensitization content, etc...)

The authors have also described quantitative and qualitative surveys, field notes, and meeting reports which lead to adjustments in the implementation. All these element should be described in the method section. 

Other important missing information concerns the description of the skin snip method, how patients were informed, how the census was conducted, how adverse event where recorded, and the origin of the drug (doxycycline).

Lines 94-96 This sentence is not clear; it is suggested to change it as follows: This intervention was supported by field monitoring, detailed notes, and a mid-point qualitative evaluation. Please clarify what it is meant by ‘mid-point qualitative evaluation’

Lines 96-97 It is suggested to change it as follows: In addition, partners on the field provided reports and meeting minutes summarizing encountered challenges and adjustments made during the implementation. Please provide further details about these challenges and adjustments or better, please provide a summary in the supplementary material. Otherwise, consider removing this sentence.

Line 98 Could you please elaborate on what was done to mobilize and sensitize the communities? Could you also elaborate on the census?

Line 99 Are the eligibility criterion only ‘above 8 years of age, ‘non pregnant nor breastfeeding’, and ‘without severe illnesses?

Lines 108-110 This sentence is not very clear, it is suggested to modify it as follows: ‘A logistic regression model was used to compare the screening participation, the positivity proportion, the treatment starts and the proportion of participant completing the treatment course in both interventions. This analysis was refined by considering subgroups such as sex, age, community, length of stay in the community, and tribe’. Please provide additional information on the type of regression models used, the parameters, and the assumptions used, and lastly summarize the identified trends. This information could be provided as supplementary material or in the result section.

Lines 110-112 As above, it seems that the authors want to put emphasis on a process that was put in place to monitor the field intervention. Perhaps, instead of mentioning that meetings and minutes have taken place, it would be useful to describe when meetings were organized, what is their scope and what were the outcomes. Concretely, it would be useful to understand what field observations were made, and which adjustments were made.

Lines 118-131 The authors should explain what the focus group discussion and the Key Informant Interviews are. It would be also useful to understand in detail how these groups are formed, how the discussion is structured, and what questions were asked. The link with table 1 is also not clear. A legend for table 1 should be provided.

Reviewer #2: The objectives of the manuscript are clearly stated, including the intention to describe the findings from field operations, what has been used and the consequent utilization of quality-quantitative methods to assess the intervention acceptance, uptake and barriers by the target population.

Reviewer #3: -Are the objectives of the study clearly articulated with a clear testable hypothesis stated?

could be more clear what the main objectives of this paper is - there is a companion paper in press which might also be reporting some overlapping data. The paper reference both 'effectiveness' ie the parasitological efficacy of implementing doxycycline in reducing skin mf and 'acceptability' ie adherence of the alternative strategy in the communities trialled. But there is no real analysis of the effectiveness. Is this because it is reported elsewhere? The authors need to clarify

-Is the study design appropriate to address the stated objectives?

yes mixed methods quant and qualitative methods to capture 'acceptability' varying between different community groups.

-Is the population clearly described and appropriate for the hypothesis being tested?

Need more details in the flow chart to match all the various % calculations discussed in the results which I found very confusing and hard to follow.

I am also really not clear how this intervention is separated geopgraphically from other alternative strategies implemented (twice annual CDTi, vector control) which are mentioned in the text. Much more clarity and information needs to be provided to determine whether there is a risk that overlapping alternative strategies are not influencing either 'effectiveness' or 'acceptability' of TTd?

-Is the sample size sufficient to ensure adequate power to address the hypothesis being tested?

a generally moderate to large sample size for the multivariate logistic regression The sample size for FGD and KII were not justified 

-Were correct statistical analysis used to support conclusions?

yes although the summary statistics were very confusing and attention needs to be paid to the flow chart to clarify this.

Two important analyses I found were missing:

1. change in mf prevalence baseline to follow up ie 'effectiveness' (no statistics - is this because this is being reported elsewhere?)

2. analysis that round two participant enrollment was SIGNIFICANTLY higher than round one.

-Are there concerns about ethical or regulatory requirements being met?

YES - it was not clear whether pregnancy test was done or the requirement for this in ethics. This is standard for Test and Treat with a class D drug.

**Results**

-Does the analysis presented match the analysis plan?

-Are the results clearly and completely presented?

-Are the figures (Tables, Images) of sufficient quality for clarity?

Reviewer #1: The result section should be improved:

Figure 1 needs to be corrected (see below). A legend describing tables 2 and 3 should be added, in addition, a detailed explanation should be included in the text. 

As in the method section, qualitative and quantitative assessments are described, however, there is no description of these assessments nor details on how they have been analyzed.

As commented in the method section, the regression models and the statistical analysis are not adequately described. It is essential in this section to clearly explain what statistical test and what regression model was used to analyze the data and how this analysis is liked with the conclusions. 

Figure 1

Please provide a legend summarizing the eligibility criterion. Please clarify why residents were not tested or treated. Please define the dropout. Please check the numbers: there is an inconsistency in line 9, a missing number in line 8, and line 20 the dropout should be written as 8 and 9 (not 08 and 09), finally it seems that a line was deleted (line 21).

Line 143 Please clarify the number of the intervention. I guess it is intervention 1. 

Line 144 Please clarify the eligibility criterion or refer to the method section. 

Line 145 what test? Skin snip?

Line 150-151 Please explain the reason why not all patients were not tested. Why do the authors report different percentages of participation based on the total and the eligible population? It is not obvious why this is relevant.

Line 156 Please explain what quantitative assessment and what test.

Line 164 as above, what test?

Line s165-168 What multiple regression model was used? This should be detailed in the method section and summarized in the result section. In table 3 the authors need to explain what the column's odds mean. Is it the probability to participate in a test? And what test? How are the odds of positivity defined? Is it mf in the skin? Or in the eyes? Or both? Please clarify what it is meant by ref. Same for the treatment start, what do odds mean in this case? Please provide an extensive legend and an appropriate explanation in the text.

Line 171 Please defined the qualitative assessment. How was it done? How was it elaborated? 

Lines 186-188 Please describe what changes were made. What was the impact of these changes on the second intervention compared to the first?

Lines 195-196 Please defined the qualitative assessment. 

Lines 204-208 It is not described how the census was conducted. Please explain what statistical analysis was conducted. As above, please describe the qualitative.

Lines 299 What is the method to report side effects?

Reviewer #2: (No Response)

Reviewer #3: -Does the analysis presented match the analysis plan?

yes for acceptability evaluations apart from some instances where quantitative significant differences are not then discussed or followed up via FGD / KII (e.g. inter-community variation in acceptability)

Whereas the parasitological reduction in prevalence from 35% to 12% is not presented at all and only mentioned in the abstract. Thus, I am only guessing that the former aspect is reported elsewhere? If so, the authors need to re-write the paper to indicate that is is only focussing on acceptability with the efficacy of TTd reported elsewhere etc.

-Are the results clearly and completely presented?

The table is OK - I can follow this. But I would like the flow chart to reflect the various summary % discussed in the results because this is extremely hard to follow.

-Are the figures (Tables, Images) of sufficient quality for clarity?

Yes.

**Conclusions**

-Are the conclusions supported by the data presented?

-Are the limitations of analysis clearly described?

-Do the authors discuss how these data can be helpful to advance our understanding of the topic under study?

-Is public health relevance addressed?

Reviewer #1: The conclusion seems reasonable. However, the method and result sections should be improved and aligned to support these conclusions. Currently, the manuscript lacks coherence and it isn't easy to link all sections.

Reviewer #2: Conclusions and results well presented, including public health implications and programatic relevance for future interventions. One recommendation would be to clearly state that data record can be vulnerable to error as it entails data entry to excel sheets from paper, which is prone to human error.

Reviewer #3: -Are the conclusions supported by the data presented?

I found the conclusions that doxycycline should be implemented with ivermectin as part of CDTi missing the point that this is a curative treatment which would not need to be delivered annually!

Other conclusions on general implementation strategy are logical and supported by data.

-Are the limitations of analysis clearly described?

certain limitations are alluded to 'delays in doxycycline treatment after testing' are alluded to but not fully explained.

Is this population affected by other alternative strategies? this confounding limitation is not discussed.

Is this area of ongoing transmission? 

I have already mentioned that parasitological analyses are missing and so are limitations around the drop in prevalence indicated in the abstract

-Do the authors discuss how these data can be helpful to advance our understanding of the topic under study?

yes

-Is public health relevance addressed?

yes

**Editorial and Data Presentation Modifications?**

Reviewer #1: (No Response)

Reviewer #2: (No Response)

Reviewer #3: I have commented that the results text discussing eligibility, testing and treatment in rounds 1 and 2 really are very confusing to follow. A re-drawing of the flow diagram would help support the text. If the text could be split into 

census

enrollment 

testing 

treatment commencement / completion for both rounds 1 and 2 that would aid the thread of the results. This should be mirrored by the flow chart.

I also think the authors need to emphasise whether parasitological data on skin mf prevalence are to be discussed eleswhere and focus on acceptability of this approach OR to fully included these analyses here. It is an impressive reduction in skin mf prevalence only very tersely mentioned in the abstract! Clearly the intervention has been of benefit.

**Summary and General Comments**

Reviewer #1: The content of this work illustrates well the difficulties to implement a field intervention, the differences between the communities, and the importance to engage with them for a successful intervention. This topic certainly deserves to be published in a scientific journal. However, in its current state the method, the presentation of the results, and the general coherence of the manuscript is not sufficient.

In addition, the authors should include, at least in the supplementary material, all the results of the survey, meetings notes, a summary of discussions, and the material used for the sensitization and to inform patients. This includes all the blank forms, field forms, and all the modifications that occurred.

The authors could also have elaborate more on the feasibility to generalize community-directed distributions of doxycycline, the side effects of doxycycline, the pro, and cons compare to community-directed distributions of ivermectin, and lastly the anticipated efficacy.

The authors have described some feedback given by the treated communities. It would be interesting to elaborate more on the suitability of community-directed distributions of doxycycline in some communities over others.

Additional comments:

The abstract is difficult to follow and needs significant reworking. It should be shortened, to summarize the objective of this work and the main conclusions.

The statement of the Financial Disclosure is quite surprising (This work is funded by Sightsavers. The funder had no role in study design, data collection, and analysis, decision to publish, or preparation of the manuscript) when we see that the first author of the manuscript is affiliated to the Sightsavers office. This contradiction must be clarified.

Reviewer #2: The study is relevant as address a programatic approach to a major neglected condition affecting vulnerable population. 

The method is clear, the study population also clearly stated. Regarding the study population, it would be relevant to add some overall comments and a brief description of main features of the semi nomadic groups covered in the study. As there are clear programatic recommendations targeting this group because of need to design culturally appropriate messages, adapt strategies to sensitize with tailored community messages, I believe the readers would benefit from further information of the group composition that might support findings as well as recommendations for future interventions.

Reviewer #3: There are many issues precluding immediate publication

There is an issue as to what the extent of this manuscript is discussing 'effectiveness' v 'acceptability' and a potential for duplicate data being published in parallel elsewhere - the authors need to be much more transparent regarding this or wait until the in press manuscript is in the public domain.

I have raised an issue of ethics regarding what was done to ensure pregnant women were not offered doxycycline.

The mode of action of doxycycline, its curative activity and thus its distinctiveness compared with the microfilaricide, ivermectin, needs much more attention in the manuscript. This is a novel approach and so the implementation required is radically different.

I have provided an annotated manuscript to help the authors.

Finally, I am aware that two senior African scientists were involved in conceptualisation and implementation of this TTd study (Didier Bakajika & Samuel Wanji) are not on the authorship list - this may be an oversight but needs justifiying. Perhaps they are acknowledged in the 'in press' article that I have not had access to.

Further I am aware that this protocol was developed by the COUNTDOWN Consortium / Liverpool School of Tropical Medicine and that SightSavers consulted with COUNTDOWN regarding implementing this protocol prior to publication of the COUNTDOWN TTd protocol in 2019 (ie before it was in the public domain). Therefore it is proper that COUNTDOWN is appropriately recognised in the acknowledgements section for sharing this information and providing advice.

PLOS authors have the option to publish the peer review history of their article (what does this mean?). If published, this will include your full peer review and any attached files.

Reviewer #1: No

Reviewer #2: Yes: Carolina Batista

Reviewer #3: No
---

## [Decision Letter · Decision Letter 1]

13 Feb 2023

Dear Dr. Nditanchou,

Thank you very much for submitting your manuscript "Acceptability of test and treat with doxycycline against Onchocerciasis in an area of persistent transmission in Massangam Health" for consideration at PLOS Neglected Tropical Diseases. As with all papers reviewed by the journal, your manuscript was reviewed by members of the editorial board and by several independent reviewers. The reviewers appreciated the attention to an important topic. Based on the reviews, we are likely to accept this manuscript for publication, providing that you modify the manuscript according to the review recommendations. 

Sincerely,

Maria Angeles Gómez-Morales, PhD

Academic Editor

Eva Clark

Section Editor

Reviewer's Responses to Questions

**Key Review Criteria Required for Acceptance?**

**Methods**

-Are the objectives of the study clearly articulated with a clear testable hypothesis stated?

-Is the study design appropriate to address the stated objectives?

-Is the population clearly described and appropriate for the hypothesis being tested?

-Is the sample size sufficient to ensure adequate power to address the hypothesis being tested?

-Were correct statistical analysis used to support conclusions?

-Are there concerns about ethical or regulatory requirements being met?

Reviewer #1: Congratulations to the authors for improving the manuscript and addressing comments from the reviewers. The method section is now clear and the statistical method is well described.

Reviewer #3: OK

**Results**

-Does the analysis presented match the analysis plan?

-Are the results clearly and completely presented?

-Are the figures (Tables, Images) of sufficient quality for clarity?

Reviewer #1: Results are well presented

Reviewer #3: OK

**Conclusions**

-Are the conclusions supported by the data presented?

-Are the limitations of analysis clearly described?

-Do the authors discuss how these data can be helpful to advance our understanding of the topic under study?

-Is public health relevance addressed?

Reviewer #1: The structure supports more clearly the conclusions.

Reviewer #3: OK

**Editorial and Data Presentation Modifications?**

Reviewer #1: line 153: the web link is repeated twice

line 706: I agree that alternative diagnostics tools should be developed for Onchocerciasis. And, in agreement with the WHO guideline, the authors have suggested the use of a serological diagnostic tool based on the OV16 antigen. 

It was however recently shown in Cameroon (10.1371/journal.pntd.0010380) that parasitological (skin biopsy) and immunological (OV16 antigen) parameters are in moderate agreement. I would therefore suggest mentioning the development of new diagnostics tools in general instead of insisting on OV16. Alternatively, the authors could provide a few additional examples of possible diagnostic techniques: the Multiplex (10.1186/s13071-019-3824-x), LAMP, etc... 

As there are several acronyms in this article, the authors may consider including a small glossary.

Reviewer #3: nonw

**Summary and General Comments**

Reviewer #1: Congratulations to the authors for this work.

Reviewer #3: the authors have addressed my major comments and restricted their data and conclusions to acceptability of implementing a combination of alternative strategies for the elimination of onchocerciasis in a persistant community focus in West Cameroon. The paper is now sufficiently clear for publication, in my opinion.

PLOS authors have the option to publish the peer review history of their article (what does this mean?). If published, this will include your full peer review and any attached files.

Reviewer #1: No

Reviewer #3: No

Figure Files:

Data Requirements:

Reproducibility:

References

---

## [Editor Report · Decision Letter 2]

21 Feb 2023

Dear Dr NDITANCHOU,

We are pleased to inform you that your manuscript 'Acceptability of test and treat with doxycycline against Onchocerciasis in an area of persistent transmission in Massangam Health' has been provisionally accepted for publication in PLOS Neglected Tropical Diseases.

Best regards,

Maria Angeles Gómez-Morales, PhD

Academic Editor

Eva Clark

Section Editor

As there are several acronyms in this article, please provide a small glossary, i.e. a list of the terms used in the manuscript in alphabetical order with the corresponding definitions.

---

## [Editor Report · Acceptance letter]

22 Mar 2023

Dear Dr NDITANCHOU,

We are delighted to inform you that your manuscript, "Acceptability of test and treat with doxycycline against Onchocerciasis in an area of persistent transmission in Massangam Health District," has been formally accepted for publication in PLOS Neglected Tropical Diseases.

Best regards,

Shaden Kamhawi

co-Editor-in-Chief

Paul Brindley

co-Editor-in-Chief
